# Can We Design a Nogo Receptor-Dependent Cellular Therapy to Target MS?

**DOI:** 10.3390/cells8010001

**Published:** 2018-12-20

**Authors:** Min Joung Kim, Jung Hee Kang, Paschalis Theotokis, Nikolaos Grigoriadis, Steven Petratos

**Affiliations:** 1Department of Neuroscience, Central Clinical School, Monash University, Prahran, VIC 3004, Australia; erica.kim@monash.edu (M.J.K.); jungheekang427@gmail.com (J.H.K.); 2B’ Department of Neurology, Laboratory of Experimental Neurology and Neuroimmunology, AHEPA University Hospital, Stilponos Kiriakides str. 1, 54636 Thessaloniki, Macedonia, Greece; ptheotokis@gmail.com (P.T.); grigoria@med.auth.gr (N.G.)

**Keywords:** Nogo-A, Nogo receptor, multiple sclerosis, axonal dystrophy, inflammatory demyelination, myelin debris, remyelination

## Abstract

The current landscape of therapeutics designed to treat multiple sclerosis (MS) and its pathological sequelae is saturated with drugs that modify disease course and limit relapse rates. While these small molecules and biologicals are producing profound benefits to patients with reductions in annualized relapse rates, the repair or reversal of demyelinated lesions with or without axonal damage, remains the principle unmet need for progressive forms of the disease. Targeting the extracellular pathological milieu and the signaling mechanisms that drive neurodegeneration are potential means to achieve neuroprotection and/or repair in the central nervous system of progressive MS patients. The Nogo-A receptor-dependent signaling mechanism has raised considerable interest in neurological disease paradigms since it can promulgate axonal transport deficits, further demyelination, and extant axonal dystrophy, thereby limiting remyelination. If specific therapeutic regimes could be devised to directly clear the Nogo-A-enriched myelin debris in an expedited manner, it may provide the necessary CNS environment for neurorepair to become a clinical reality. The current review outlines novel means to achieve neurorepair with biologicals that may be directed to sites of active demyelination.

## 1. Introduction

Multiple sclerosis (MS) is a neurodegenerative disorder predominantly affecting the female Caucasian population with median incidence less than 40 years of age. It is estimated that over 2.4 million people are living with MS worldwide [1]. Its chief clinical manifestation is demyelination in the central nervous system (CNS) including the brain, spinal cord, and optic nerve. The pathogenesis of MS manifests as an acute and chronic inflammatory mediated demyelination of the CNS [1]. Patients with MS can exhibit heterogeneous neurological deficits including cognitive and motor impairment such as visual disturbances, tremor, muscle weakness, numbness, tingling, and significant fatigue [1]. While currently available treatments are based on immunomodulatory therapies, that affect the inflammation status and sequelae of the patients diagnosed with MS, with specific or indirect mechanisms of action to suppress T and B cells, monocytes, and the innate immune cells, there still remains a significant unmet medical need to prevent longstanding demyelination and promote neural repair through remyelination. In this review, the pathogenesis of MS will be discussed thoroughly including mechanistic aspects. Furthermore, immunomodulatory drugs that are clinically available and under clinical trial will be reviewed and a novel intervention of delivering a therapeutic protein to enhance recovery in MS will be discussed with existing evidence.

## 2. Clinical Features of MS

The disease pattern of MS is unpredictable and divergent amongst patients. This unpredictability is due to the disseminated lesions that can propagate throughout the CNS at any time. This clinical feature has been attributed to heterogeneity of the disease and the difficulty of early diagnosis defining clinically isolated syndrome (CIS), treatment, and disease co-morbidities. The diagnosis of MS is based on the current McDonald criteria magnetic resonance imaging (MRI) scan of lesions along with the clinical profile [2]. Each subtype of MS exhibits a distinct course of clinical progression. Relapsing-remitting MS (RRMS) is the most common form of MS, which is defined by acute episodes of worsening of neurological function followed by a variable degree of neurological and functional recovery with a stable course between attacks [3]. The majority of RRMS patients may progress into secondary progressive MS (SPMS), which involves steady progression between a pattern of relapses. Primary progressive multiple sclerosis (PPMS) affects 10–15% of the overall MS patient population. The disease progression of PPMS manifests as continuously worsening neurological symptoms without remissions and accumulation of MS lesions over time and age with the progressive destruction of axon-myelin integrity and failed remyelination that may ultimately lead to irreversible neurodegeneration such as CNS atrophy and axonal damage, of which the outcome is permanent disability [3]. The least occurring form of MS, is progressive-relapsing MS (PRMS), involving recurrent phases of acute inflammatory attacks that may be irreversible during the remission period. The newly revised clinical phenotypes of MS include recommendations for MS disease course nomenclature [4]. These two phenotypes include radiologically isolated syndrome (RIS) along with clinically isolated syndrome (CIS). The RIS classification was included by the advisory committee commissioned by the National Multiple Sclerosis Society (NMSS) because it was felt that due to the advancement in MRI technology, a greater frequency in patients without clinical signs or symptoms of MS were being identified under routine MRI. This included incidental findings highly suggestive of demyelination [5]. These patients were found to have a five-year risk of 34% for acute or progressive first-up clinical events [5]. Furthermore, the committee approved CIS as a MS disease course since it did not meet the definition of dissemination in time for diagnosis of clinically defined MS [6]. All of these MS phenotypes show our increasing understanding of the pathologies exhibited by individuals living with this severe neurological condition and are assisting our research endeavors towards treatment and patient management of the pathological features and their complications.

The well-established clinical and research dogma is that inflammation plays a key role in MS, where peripheral immune cells infiltrate into the CNS, through a ‘leaky’ blood–brain barrier, to primarily target myelin antigens, precipitating the demyelinating pathology so commonly attributed as the pathogenesis. However, evidence exists that implicate axonal damage and failure of remyelination as major determinants of profound neurological deficit in progressive MS patients [7,8,9,10]. Whether these neural targets are indeed primary or secondary in MS lesions, separated by time and space, is still being debated.

## 3. Pathological Features of MS

The pathological features of MS can be divided into two broad aspects; (i) inflammation driven by immune cells from the periphery, and (ii) demyelination leading to axonal pathology, which will be discussed in detail.

### 3.1. Inflammation Driven by Peripheral Immune Infiltrates

Historically, MS is considered as an autoimmune disorder, with purported autoreactive lymphocytes—predominantly T cells—as the inducer of the MS pathogenesis. The immunopathology of MS has been extensively reviewed elsewhere and will not be covered here (for review, see [1]). Peripheral immune cells infiltrate into the CNS through a ‘leaky’ blood–brain barrier, to primarily target myelin antigens, precipitating in demyelinating pathology, so commonly attributed as the pathogenesis. The peripheral immune infiltrates in MS are composed mainly of lymphocytes, including CD8+ T cells, and fewer CD4+ T cells and B cells [11]. The myelin-specific CD8+ T cells particularly may play a greater role in propagating MS pathology as it is believed these T cells exert effector functions that cause progression of neural injury to exacerbate the MS pathological sequelae [11]. Further evidence suggests that elevated CD8+ T cell infiltrates have been highly correlated with demyelination and axonal injury in cortical regions of the MS brain [12,13]. Moreover, CD8+ T cells may contribute to oligodendrocyte death and axonal impairment via CD8+ T cell-mediated cytotoxicity since these cells have been observed within lesions [14]. These data suggest that these inflammatory responses exacerbate the pathogenesis of MS leading to neural destruction.

Upon T cell activation, the inflammatory cascade is initiated, and inflammatory mediators are elevated. Macrophages and microglia may acquire an M1 phenotype potentiating neuro-destructive effects by releasing pro-inflammatory cytokines such as interleukin (IL)-1β, tumor necrosis factor (TNF)-α, and IL-6 (for review, see [15]). Subsequently, their phagocytic activity on myelin and axonal structure are activated and as a consequence, the myelin sheath is stripped away from cluster of axons, leading to their denudement followed by the eventual transection and Wallerian degeneration, causing severe neurological deficits. These axons are vulnerable to degeneration, beginning with impaired axonal transport. However, macrophages and/or microglial cells exhibiting a M2 phenotype may potentiate neuroprotective and neurotrophic mechanisms by releasing anti-inflammatory cytokines, such as IL-4 and IL-13 (for review, see [16]).

These M2 tissue-specific monocytic lineage cells may enhance the clearance of myelin debris as it is recognized that during this degenerative state, myelin proteins can inhibit neural repair [17]. The role of myelin-clearing phagocytic macrophages or CNS-resident microglia in local immunosuppression has also been implicated in recent investigations [17]. Studies have revealed the presence of myelin-laden macrophages releasing anti-inflammatory molecules in demyelinating MS lesions and the suppression or switching of detrimental M1 phenotypic macrophage/microglial cells to M2 anti-inflammatory phenotypes, followed by myelin phagocytosis [18]. The switching between M1 and M2 phenotypes is a microenvironment-dependent and tightly regulated process essential for tissue homeostasis [18]. However, this nomenclature is not entirely accurate since it has been defined that once monocytic cells enter the CNS compartment and become inflammatory macrophages, they express markers of both M1 and M2 but may show differing pathophysiological sequelae [19]. Indeed, recent nomenclature of activated macrophages that either produce tissue damage (the pro-inflammatory, damage associated macrophages or DAMs) or anti-inflammatory alternatively-activated macrophages, suggest that identification of macrophage phenotype can only be assessed through transcriptomic analysis, illustrating their responsiveness to tissue ligands which govern their physiological profile [20]. Despite these recent profiling criteria, histopathological observations continue to report that optimizing the microenvironment toward neuroprotection and neurorepair may be facilitated by M2-macrophages and microglia through myelin debris clearance, highlighting an alternative endogenous therapeutic option to resolve MS plaques [21,22].

Analysis of post-mortem progressive MS brain tissue has shown a higher number of B cells found localized to the meninges along with oligoclonal bands appearing in the cerebral spinal fluid (CSF) of MS patients [23,24]. Traditionally, B cell responses in MS were considered to be secondary to, and dependent on T cell activation which has been re-evaluated in B cell deficient mice following inflammatory-mediated demyelination, that exhibited a reduction in the degree of demyelination [25]. It has been demonstrated that short-lived plasma B cells are present in the CSF of MS patients, producing specific antibodies that can target CNS myelin components [26,27]. Clones of antibody-secreting B cells may expand and mature by forming germinal centers within the meninges of the brain, which may activate or re-activate CNS-localized B cells. Recent studies have discovered B cell follicle-like structures in the CSF, meninges and cortical grey matter of progressive MS patients, propagating cycles of inflammatory demyelination in the cortical structure and exacerbating progressive neurodegeneration in MS [23,28]. Furthermore, it has been shown that MS patients display an increased level of B cell activating factor (BAFF) [also known as tumor necrosis factor ligand superfamily member] in lesions, which is an important mediator for B cell survival and growth [29]. The upregulation in macrophages and T cells may well be due to the expression of BAFF, highlighting the importance of B cells pertaining to demyelination and axonal damage. It is notable that astrocytes may be a non-immune source of BAFF since histologically there is co-localization of BAFF and astrocytes in close proximity to BAFF receptor expressing immune cells during neuroinflammation [30]. Collectively, these inflammatory elements instigate the exacerbation of MS, leading to the ultimate destruction of myelin sheaths, oligodendrocytes, and axonal degeneration.

### 3.2. Demyelinating Plaques

Demyelination is one of the hallmarks of MS, which is defined as the destruction of the myelin sheath around axons mediated by inflammatory infiltrates of the CNS, and subsequently leading to axonal degeneration. As the immune cells infiltrate into the CNS from the periphery, they may target myelin antigens including CNPase, myelin-oligodendrocyte glycoprotein (MOG), proteolipid protein (PLP), and myelin basic protein (MBP) although such a hypothesis remains purely extrapolated for autoimmune models of the disease. The multiple focal areas of demyelination within the CNS are called plaques or lesions [31].

Distinctive phases of demyelinating activity may define types of plaques such as acute or chronic, and active or inactive. Acute active plaques are more prevalent in acute and RRMS patients, which are characterized by hypercellular demyelinated plaques with the presence of myelin-laden macrophages (Figure 1). The degradation of minor myelin proteins such as CNPase, and MAG is relatively rapid, which occurs within one to three days, whereas the degradation of the larger, more abundant and hydrophobic major myelin proteins such as PLP and MBP is slower and may persist in lesions for up to 10 days [31]. An inactive plaque is defined as the presence of infiltrated macrophages but a lack of myelin debris (Figure 1) [32,33]. Chronic plaques are more predominant than active plaques in progressive MS patients. These plaques are demarcated demyelinated lesions where their core is a hypocellular pool of myelin-laden macrophages and are characterized by activated microglia surrounding the inactive core (Figure 1) [31,32,33]. Chronic inactive plaques are completely demyelinated, hypocellular, and consist of the substantial loss of axons and oligodendrocytes, along with the presence of gliosis, and minor immune infiltrates (Figure 1) [31,32,33].

In the case of arrested inflammatory insult at the early phase of demyelination, plaques may be partially remyelinated, pathologically defined as shadow plaques [31,32,33]. Such lesions may have their associated normal appearing white matter (NAWM) showing milder pathology [31,32,33]. However, it has been determined that shadow plaques are more susceptible to second-hit inflammatory insults than the NAWM [34]. In a healthy environment, remyelination may occur spontaneously following the insult to the CNS by the endogenous stem or precursor cells with the capability to repair demyelinated axons and restore functional loss [35]. However, this often fails in MS brains possibly due to prolonged cycles of inflammation and a cumulative lesion burden, or failure of resident OPCs to differentiate into myelinating oligodendrocytes or glial scar formation that creates a barrier between oligodendrocytes, and axons in progressive MS patients [35].

### 3.3. Mechanisms of Axonal Pathology

For many years now, the debate over the two opposing paradigms of axo-glial disintegration during the pathogenesis of MS, has continued without resolution. There exists archival histopathological evidence for both the ‘outside-in’ or ‘inside-out’ etiopathogenesis that may govern lesional activity (demonstrated in Figure 1) [36]. Despite valid arguments posited for both mechanisms of neural damage, the current evidence suggests that these can be operative simultaneously. The first paradigm defines the ‘outside-in’ mechanism of axonopathy and demyelination, whereby the peripheral immune infiltrate-mediated inflammation is the initiating factor for primary demyelination, thereby potentiating secondary axonal degeneration [37]. Moreover, primary oligodendrocyte dystrophy has been observed in human MS demyelinated lesions, that are histopathological patterns of neurodegeneration reminiscent of toxic events and/or viral infection and correlated to acute progressive disease [33]. These defined immunopathological pattern III and IV MS lesions suggest that progressive demyelination effecting oligodendrocyte viability from an ‘inside-out’ manner, may be induced by damaging structural myelin and oligodendrocyte proteins such as MAG that are important to sustain the intimate relationship shared between the oligodendrocyte and the axons they ensheath (Figure 1) [33].

In contrast to this hypothesis, post-mortem analysis of spinal cord sections from a patient with acute MS lesions has shown that the axonal injury was present in normal-appearing white matter (NAWM), with normal immunostaining for myelin [38]. Furthermore, it has been found that the axonal density was also reduced in NAWM of MS patients [9]. This emerging evidence suggests that an ‘inside-out’ mechanism may be operative in the brain lesions of individuals with MS, exhibiting primary axonal degeneration propagating a secondary demyelination [39]. Further evidence for this disease paradigm derives from our emerging understanding in metabolic dysfunction in myelinating oligodendrocytes driving neurodegenerative responses [40]. Indeed, altered oligodendroglial cell metabolism may be occurring in the CNS of individuals during progressive MS [41]. In a groundbreaking study, it was eventually identified that the acute downregulation of the lactate plasma membrane transporter, monocarboxylate transporter 1 (MCT1), in myelinating oligodendrocytes within the optic nerves of mice, produced substantial axonal degeneration [40]. These data implicated MCT1 for the first time as a potent transporter in oligodendrocytes regulating the metabolism and hence, structural integrity of the axons that they myelinate, through a lactate-dependent nourishment of neurons [40]. This evidence may suggest an ‘inside-out’ mechanism driving neurodegeneration during MS progression but remains to be validated in variable MS lesions and models of the disease [41,42].

It is likely that both mechanisms are operative with either primary or secondary axonal degeneration culminating in neurodegeneration. It is well-defined that axonal degeneration is initiated at the Node of Ranvier, which is a characteristic event not restricted only to the pathogenesis of MS, but other neurodegenerative conditions including spinal cord injury and glaucoma [43,44,45]. Axonal degeneration involves macrophage activity and substantial axonal swelling occurs over time as macrophages strip away the myelin and engulf myelin debris, causing significant axonal damage such as axon varicosities and dystrophic axons [37]. The presence of substantial myelin debris, extracellular to those macrophages, express myelin-associated inhibitory factors which inhibit the recruitment of OPCs to the lesion and subsequently the denuded axons are more vulnerable to degeneration. Chief among the myelin-associated inhibitory factors is Nogo-A. The pathological mechanism governing this inhibitory factor associated signaling will be discussed, which may be a potential therapeutic target to enhance the clearance of myelin debris and enable the remyelination of denuded axons by resident OPCs (discussed below).

## 4. Nogo-A/NgR-Dependent Mechanisms Governing Neuroinflammation and MS Pathology

### 4.1. Myelin Associated Inhibitory Factors (MAIFs)

Myelin associated inhibitory factors (MAIFs) consist of Nogo-A, myelin-associated glycoprotein (MAG), and oligodendrocyte-myelin glycoprotein (OMgp) and are expressed on myelin and oligodendrocyte membranes, limiting axonal regeneration in experimental models of neurotrauma and axonal damage [46]. These MAIFs may also potentiate axonal degeneration in the animal model of MS (for review see [46]), as observed in experimental autoimmune encephalomyelitis (EAE) [47]. As hypothesized in MS and models of demyelination, it is believed macrophages attack myelin at the paranodal regions stripping away its compact membranes, rich in these MAIFs. Extracellular myelin debris are generated from the stripped sheaths of membrane, ready to be cleared by activated macrophages through phagocytosis [48]. A plausible explanation for remyelination failure in progressive MS may relate to the co-activation of MAIFs within extracellular myelin debris acting as potent endogenous inhibitory cues which block the recruitment and maturation of OPCs around the lesion to repair damaged myelin [49,50]. It has been suggested that MAIFs inhibit axonal outgrowth and extension of growth cones through the activation of the small GTPase RhoA regulating actin and tubulin disassembly via a signaling pathway which may be kinase-dependent (Figure 2) [46,51]. This MAIF ligand-dependent signaling, is mediated by a tripartite receptor complex of NgR1, neurotrophin receptor (p75^NTR^), and LINGO-1, expressed on axons (Figure 2) [52,53,54].

Chief amongst the MAIFs, it has been identified that Nogo-A has two functional domains that contribute to the inhibition of neurite growth; amino-Nogo which is seen either in a cis co-formation localized to the cytoplasmic side of the cell membrane and Nogo-66, which consists of 66 amino acid residues at the C-terminus, extracellularly (Figure 2) [55]. Nogo-66 is followed by two hydrophobic domains and it is more well documented that Nogo-66 can potentiate a potent inhibitory effect during neurite growth through its strong binding to its cognate receptor, Nogo receptor (NgR), predominantly expressed on neurons (Figure 2) [56,57]. However, signaling commonly occurs through its co-receptors; p75, LINGO-1, and TROY [53,54,58]. Nogo-A has been an attractive target in MS research, whereby a therapeutic passively transferred antibody, targeting Nogo-A during the course of EAE in mice, can reduce the incidence, severity and pathological sequelae, by limiting axonal damage [47]. Subsequent in vivo studies suggest anti-Nogo-A treatment can improve myelin repair and functional recovery after inflammation-mediated and lysolecitihin-induced demyelination, without altering the number of M1 macrophages and astrocytes [59].

Myelin-associated inhibitory factors (MAIFs) include MAG, Nogo-A, and OMgp are physiologically expressed upon myelin and oligodendrocyte plasma membranes. The cognate receptor for these MAIFs, Nogo receptor 1 (NgR1) can multimerize with the co-receptors LINGO-1, p75, and/or TROY to signal neurite outgrowth inhibition. Upon the activation of NgR1-dependent signaling, intracellular molecular switches, that can include RhoA-GTP, are activated and thereby the phosphorylation of the Rho-associated, coiled-coil containing protein kinase 2 (ROCK II) can occur unimpeded, with further downstream phosphorylation of the threonine 555 site on the C-terminus of the microtubule-related protein, collapsin response mediator protein 2 (CRMP2). These downstream molecular events can lead to the disassembly of microtubules potentiating the axonal retraction and/or an inhibition of Rac-GTP activity which phosphorylates Cofilin leading to actin depolymerization. Such retraction events of axonal lamellipodia at the distal end can progress throughout the axonal microtubule system causing catastrophic depolymerization and microtubule-associated protein hyperphosphorylation, commonly identified in dystrophic axons within the chronic-active MS lesion (LRR: leucine-rich repeat).

### 4.2. Nogo Receptor 1 (NgR1)-Dependent Neurobiological Events

Nogo receptor 1 (NgR1) is a transmembrane protein expressed on the surface of axons in the CNS and its key binding domain is an extracellular leucine-rich repeat domain [52]. The two other inhibitors, OMgp and MAG, also bind to NgR, although their specificity seem restricted, with MAG showing greater affinity to the NgR homologue, NgR2. However, OMgp shows exclusive binding to NgR1 [60,61,62]. NgR1 plays a central role in facilitating neurite outgrowth dynamics, mainly through an interaction with the inhibitory extracellular milieu [52]. Importantly, it may also govern the plasticity of the CNS at the axo-dendritic [63] and axo-glial levels [64]. A recent study on NgR1-knockout mice (*ngr1*^−/−^) has shown that these mice exhibit re-distribution of the paranodal Caspr protein with longer nodes of Ranvier and with evidence of elevated myelin turnover [64]. This study suggests myelin plasticity in *ngr1*^−/−^ mice, highlighting another potential role for NgR1 during development and axo-glial maintenance in the adults [64]. Furthermore, an immunomodulatory role has been implicated with recent identification of increased phagocytic activity [65] and, a suspected shift in macrophage phenotype from an M1-pathogenic to M2-neuroprotective, observed in *ngr1*^−/−^ mice, during EAE progression; however, whether this is a consequence of compensatory NgR2 or particular NgR2 homolog signaling remains to be elucidated [66].

It is well-established that axonal NgR1-dependent signaling plays a critical neurophysiological role in synaptic pruning and neural plasticity in the CNS [67,68,69]. During CNS neurodegenerative disorders and in animal models of inflammatory demyelination (as shown in EAE), the accumulation of MAIFs, are known to bind to their cognate receptor, NgR1 at nanomolar affinity [70,71], thereby facilitating the activation of NgR1-dependent signaling through the heteromerization with their co-receptors [72]. It has been demonstrated that axonal NgR1-dependent signaling is a key mediator in inhibiting axonal preservation and repair in EAE pathology, where activated RhoA-GTP/ROCK-II downstream transducers cause the phosphorylation of the downstream microtubule-associated protein (MAP), known as collapsin mediated protein (CRMP2) [47]. This key MAP is a modulator of axonal growth and vesicular transport [73]. This phosphorylation at the threonine 555 (Thr555) site can lead to axonal degeneration, potentially through the abrogation of axonal transport [36,47]. Moreover, it has been shown that mice deficient for the NgR1 (*ngr1*^−/−^) display a reduced severity of EAE disease sequelae, emphasizing the importance of investigating the molecular mechanism(s) underscoring this result [47]. These studies led to therapeutic approaches in an attempt to antagonize the NgR1-dependent signaling through the ligation of MAIFs, and may include peptide based biologicals and other therapeutics such as the NgR(310)-Fc fusion protein to bind and clear these inhibitory myelin proteins enriched in myelin debris (Figure 3) [74,75]. Given that studies using the ectodomain of NgR1 demonstrate that blocking extracellular MAIF binding can enhance neurological recovery following CNS injury through enhanced axonal regeneration in models of spinal cord injury and glaucoma, this is an exciting new paradigm that is being addressed by our research team in a new paradigm for MS therapeutics [74,75].

Another potential therapeutic option has surfaced following the identification of the neurobiologically active, cartilage acidic protein 1B, designated as the lateral olfactory tract (LOT) usher substance (LOTUS) protein, recently identified to antagonize NgR1-dependent MAIF signaling of fasciculated axons [76,77,78,79]. Recently, it was identified that the mean CSF soluble form of LOTUS (s-LOTUS) in SPMS cases was considerably reduced in concentration when compared with non-neurological disease control samples and those of RRMS cases [80]. More recent research has now identified the s-LOTUS as a binding protein for the low affinity neurotrophin receptor p75^NTR^, interfering with the potential heteromerization with NgR1 and hence, signal transduction [78]. Importantly these investigators identified that s-LOTUS blocked MAIF-induced neurite retraction events and promoted axonal regeneration following a crush optic nerve injury in the mouse [78]. These data implicate LOTUS as a key regulator of NgR1-dependent signaling during neuroinflammatory-mediated axonal damage with the potential to enhance axonal regrowth. Whether this endogenous protein can be modulated or artificially increased in MS remains to be established.

### 4.3. Axonal Transport Deficits

One of the key initiators of neuronal health is the physiological regulation of fast or slow axonal transport of neurotransmitters and various proteins (and even organelles) central to either growth, plasticity or maintenance of axo-dendritic interaction [81]. If this transport system slows or stalls, then remote or distal axonal connections will be starved of vital proteins and structures that keep the integrity of neuronal synapses in harmonization [81]. As mentioned earlier, when axonal NgR1-dependent signaling is amplified through the activity of RhoA-GTP/ROCK-II and CRMP2 phosphorylation, axonal growth and maintenance can be impacted, yet how these deleterious (or otherwise) effects elicit vesicular transport blockade are not exactly clear [73]. However, it has been reported that phosphorylation of CRMP2 can lead to axonal transport dysfunction [36,47], highlighted by the fact that the over-expression of the dominant negative form of CRMP2 abrogates hippocampal neuron polarization and no axons can be formed [82]. Whereas, the overexpression of a constitutively active form of CRMP2 generates multiple axons in these primary hippocampal neurons, suggesting a chief anterograde transport role for this microtubule associated protein. How this physiological feature relates to axonal transport regulation, in particular within the diseased neuron is yet to be elucidated but evidence now has linked the post-translational modification of CRMP2, either via phosphorylation or cleavage, with kinesin-dependent anterograde axonal transport [83].

The kinesins are a family of motor proteins found in eukaryotic cells that regulate meiosis, mitosis, or the movement of organelles and vesicular cargo in an anterograde fashion in axons. Previously, it has been reported that an rs10492972[C] variant in the KIF1B gene (translating some of the kinesin superfamily proteins, important for synaptic vesicle and mitochondrial transport) was associated with MS [84]. Mutations of this gene are well-characterized in the Charcot Marie Tooth Type 2A1 disease, associated with axonopathy. However, follow-up investigations identified that the Aulchenko study was not sufficiently powered to draw these conclusions [85]. Despite this, Hares et al., 2014 demonstrated reduced mRNA expression of KIF5A, KIF1B, and KIF21B, within cortical grey matter tissue from individuals who had lived with MS. All of these key motor proteins are known to regulate axonal transport mechanisms driving vesicle cargoes containing microtubule related proteins such as phosphorylated neurofilaments, amyloid precursor protein (APP) and mitochondria [86]. The reduced protein expression of KIF5A was shown to be associated with MS brain tissue across various lesional and non-lesional areas in both grey and white matter [86]. These data may suggest the motor proteins as major players in the neurodegenerative process governing progressive MS.

Research findings from MS models have not only identified axonal swelling and dystrophy as a consequence of stalled transport but that this may be recoverable to limit degeneration if transport is reestablished [39]. In a landmark paper, two-photon imaging of in vivo spinal cord axons in the MOG_35-55_-induced EAE mouse model of MS, accurately illustrated that mitochondrial or peroxisome anterograde transport is remarkably decreased through active inflammatory lesions as well as normal appearing white matter (NAWM) [87]. Importantly, these investigators revealed that the organelle trafficking during these inflammatory events displayed continual stalling of movement with these deficits preceding structural alterations in axonal integrity [87]. These data may suggest that there exists an altered motor protein movement capacity during neuroinflammation that may be reversible if treated early prior to profound neurodegeneration.

Specific axonal transport deficits have been reported in optic nerve of EAE-induced mice observed through magnesium enhanced MRI [88], with the degree in reduced optic nerve transport associated with impaired microtubule-associated proteins, demyelination, axonal dystrophy, and inflammation. Importantly, a recent study has reported that in the cuprizone-fed mice showed reduced kinesin light chain (KLC) expression within the demyelinated hippocampus, associated with learning abnormalities, improved by the therapeutic administration of anti-LINGO 1 antibody. The data suggested that demyelination potentiates the reduction in axonal transport, potentially leading to cognitive deficits. These data are placed in greater focus when reviewing the landmark study of Lyons et al. (2009), that reported a mutation in the motor domain of *kif1b* (st43) in zebra fish produced an ectopic orientation of myelin membrane [89]. Morpholino oligonucleotides (MO) designed to block all *kif1b* isoforms disrupted MBP mRNA localization in oligodendrocytes, suggesting a central role of myelination and remyelination to be a result of KIF1B function with direct relevance in axonal transport related proteins with appropriate myelination [89]. Collectively, the data implicate kinesin function as an important regulator of axo-myelin integrity, which may underscore chronic-active MS lesion evolution.

## 5. Current Therapy Options for Progressive MS

There are a number of disease-modifying therapies available for primary and secondary progressive MS including immunomodulatory drugs and autologous hematopoietic stem cell transplantation currently in clinical trials. These are mainly purposed to suppress or ablate the patient’s immune system to slow the disease progression; however, not to reverse the neurological deficits by promoting axonal regeneration and remyelination. These therapeutics will be discussed in detail.

### 5.1. Immunomodulatory Drugs

Currently, immunomodulatory drugs are used to treat RRMS and progressive MS patients targeting or suppressing the patient’s immune system. An anti-CD20 monoclonal antibody has been developed to selectively deplete CD20+ B cells and the biological therapies have produced the exciting Rituximab, and Ocrelizumab therapeutics, and have been significantly tested in multicenter clinical trials. The first generation of anti-CD20 monoclonal antibody, Rituximab, has shown its efficacy in reducing the relapse rate in RRMS patients [90]. It is designed to target the CD20 antigen expressed on B lymphocytes from the pre-B-cells to mature B cells [91]. Subsequently, it was tested in primary progressive MS (PPMS) patients in the randomized phase 2–3 of the OLYMPUS trial [92]. Although its primary efficacy end point was not met, it was suggested that the selective depletion of B cells may affect the progression of disability, with the time to confirmed disease progression being delayed in younger patients (<51 years of age) [92]. The second generation of anti-CD20 antibody, Ocrelizumab, a humanized monoclonal antibody that selectively depletes CD20-expressing cells in the PPMS patients, was tested in the phase 3 ORATORIO clinical trial [93]. The drug was delivered by IV-injection and associated with lower rates of clinical and MRI progression, compared to placebo group [93]. Ocrelizumab (Ocrevus) has been recently approved by the U.S. Food and Drug Administration (FDA) for the treatment of RRMS and PPMS. Although extended monitoring of patients is required to determine its long-term safety and efficacy, this is the first disease-modifying drug on the market effective for the treatment of PPMS.

So far, there are no existing treatments that target the slowing of disease progression in patients with secondary progressive MS (SPMS) that follows on from RRMS. The Sphingosine 1-phosphate receptor modulator has been developed to suppress the immune system by limiting egress of lymphocytes from the lymphoid tissues. The first generation of modulator is Fingolimod, which was tested in a phase 2 trial that demonstrated an effective treatment for RRMS [94,95]. Although its safety and efficacy were assessed in PPMS patients, the anti-inflammatory effects of Fingolimod did not slow disease progression in PPMS patients [96]. This led to the development of Siponimod, a selective sphingosine 1-phosphate (S1P) receptor 1 and 5 modulator that has higher specificity towards the receptor 1 and 5 [97]. Recently, a double-blind, randomized phase 3 EXPAND trial was initiated to test the effectiveness of Siponimod, in SPMS patients [98]. The initial three-month monitoring has been shown to reduce confirmed disease progression (CDP), showing its potential for treating SPMS [98]. These immunomodulatory drugs are effective only to suppress the immune system, limiting potential immune attack, but do not reverse neurological disability. There is therefore a clear medical need for novel therapeutic approaches towards regeneration and remyelination to enhance the quality of life in people living with progressive MS.

### 5.2. Autologous Hematopoietic Stem Cell Transplantation

Although still experimental, a recent therapeutic option to limit aggressive forms of MS is autologous hematopoietic stem cell transplantation (AHSCT), which originally was clinically developed for treating hematological malignancies. It may be an alternative option provided to patients with the inclusion criteria being failure in prior conventional treatment [99]. AHSCT involves a multistep procedure including extensive chemotherapy which ablates the patient’s immune system and allows reconstitution of transplanted HSCs [99]. The transplanted HSCs may suppress inflammatory activity and may suppress and slow the disease progression of MS.

A number of clinical trials have been conducted to test the feasibility and safety of this therapeutic regimen. A phase 1 clinical study (NCT00017628) of intense immune suppressive therapy involving total body irradiation, and AHSCT was conducted on twenty-one patients with rapidly progressive MS [100]. It was concluded that the study was ineffective for patients with progressive disease and further evidence for the effectiveness and safety is required [100]. Another study conducted involving a large cohort of patients with autoimmune diseases including MS, has received AHSCT, and results have shown improvement over time with sustained remissions for more than five years in patients with severe autoimmune diseases refractory to conventional therapy. Scientists suggest the disparities in successful outcomes depend on the type of autoimmune disease, rather than the technique involved during transplant [101]. Moreover, three-year interim analysis of an ongoing, multicenter, and phase 2 clinical trial of high-dose immunosuppressive therapy with AHSCT for RRMS (i.e., HALT-MS, NCT00288626) has suggested that this therapy was effective for sustaining remission of active RRMS and was associated with improved neurological function, also few serious early complications or unexpected adverse events [102].

A recent phase 2 clinical trial (NCT01099930) conducted in Canada that was non-randomized and involved a small cohort of patients with relapses or progression, has demonstrated that AHSCT halted further progression of MS without ongoing disease-modifying drugs and formation of active lesions in conjunction with clinical improvements in MS patients [103]. Furthermore, an interim result of the MIST clinical trial (NCT00273364) showed a reduction in disability scale of RRMS patients, compared to conventional immunomodulatory drugs. A report evaluating retrospective long-term outcomes in MS patients after receiving AHSCT from 1995–2006 has suggested that approximately half of recipients remained free from neurological progression for five years post-transplant. Furthermore, the outcomes correlated to younger age, relapsing form of MS, fewer prior immunotherapies, and lower disability score prior to transplant [104]. These data support the effectiveness of AHSCT for the treatment of MS, and AHSCT may bring hope to MS patients who do not respond to conventional therapy. However, an ultimate goal of regenerating damaged CNS in these patients is still unexplored.

Extracellular myelin debris expressing myelin-associated inhibitory factors (MAIFs) which include MAG, Nogo-A, and OMgp. A soluble fragment of NgR1 utilized as a potential biological therapeutic agent; this NgR(310)ecto-Fc fusion protein contains the ligand binding domain as a decoy, blocking all three myelin inhibitors from interacting with their cognate receptor, Nogo receptor 1 (NgR1). In a pathological context, upon the binding of MAIF-NgR(310)ecto-Fc complex, multimerization of the NgR1-dependent complex is achieved, which includes the prevention of RhoA-GTP/ROCK II downstream transducer activation. Limiting ROCK II activity in neurons can prevent the phosphorylation of the downstream microtubule-associated protein, collapsin response mediator protein 2 (CRMP2), and thereby promote neuronal microtubule and neurofilament stability by phosphorylation of cofilin. The blockade of these NgR1-dependent downstream molecular events by utilizing the NgR(310)ecto-Fc decoy protein can potentially enhance neurological recovery following CNS disease or injury (LRR: leucine-rich repeat).

## 6. Potential Therapeutic Interventions for Remyelination

### 6.1. Utilizing Biological Peptide—NgR(310)ecto-Fc to Drive Neural Repair in the CNS

As mentioned earlier, a soluble fragment of NgR1 contains the ligand binding domain can function as a decoy blocking all three myelin inhibitors; Nogo-66, MAG, and OMgp (Figure 3) [56,61,105,106]. Fusion of this c-terminus peptide with the Fc portion of IgG may provide a binding site for endogenous Fcγ receptors on monocytes, which may enhance the phagocytic clearance of myelin inhibitors, as shown through in vitro studies that have demonstrated human immunoglobulins in mouse sciatic nerve exhibiting increased capacities for the clearance of lesional debris and in particular the removal of MAIFs [107].

The therapeutic potential of NgR(310)ecto-Fc has been implicated in previous studies, including the local delivery of rat NgR(310)ecto-Fc in animal models of acute and chronic thoracic spinal cord contusion [108,109,110,111], dorsal root crush injury, and ischemic stroke [112,113]. Furthermore, it has been demonstrated that upon local delivery of NgR(310)-Fc, the sprouting of uncut fibers separates from the injury site, and the increase in axonal regeneration from transected fibers due to acute blockade of MAIFs [114]. Moreover, the continuous intracerebroventricular infusion of the human NgR(310)-Fc decoy protein into rats following spinal cord contusion injury, is linked with significant neurological recovery demonstrated by locomotor recovery [75]. Given that local delivery of the NgR(310)-Fc fusion protein to several injury sites has demonstrated repair outcomes in animal models there is solid rationale to pursue this therapeutic regime within the animal model of MS, EAE. However, the lesion-specific delivery of the NgR(310)-Fc protein to all disseminated lesion sites in the context of MS pathology is such an enormous challenge due to pathological heterogeneity and a more targeted therapeutic delivery requires elucidation.

### 6.2. Hematopoietic Stem Cell Delivery of the NgR(310)ecto-Fc Therapeutic Fusion Protein

As mentioned earlier, local CNS delivery of the NgR(310)-Fc decoy protein has shown axonal regeneration post-injury; however, targeting NgR1-signaling operative at specific disseminated inflammatory demyelinating lesions is extremely challenging. Localization of inflammatory cell infiltrates in the pre- or active lesions of demyelination within the CNS is a well-known pathological characteristic observed during the disease course of MS and EAE. The trafficking of this biological peptide into the CNS of progressive MS patients may be highly restricted due to the tightly-regulated blood–brain barrier (BBB). However, immune cells such as activated T cells and macrophages, do traffic through the BBB and enter the CNS homing in on neurological lesions to produce consistent secreted protein levels that make them an attractive vehicle for therapy [115]. Moreover, a systemic NgR1-Fc dose delivery may have deleterious ‘off-target’ effects, particularly since NgR1 can regulate synaptic plasticity in the adult cerebral cortex [116].

Currently, the available immunomodulatory and anti-inflammatory therapies are able to slow, but not reverse, MS disease progression. A study targeting NgR1-dependent signaling in which anti-LINGO-1 therapy alleviated EAE progression and βAPP+ demyelinated lesions [117]. This has now progressed to a phase IIb AFFINITY trial (NCT03222973). However, the therapeutic effect of anti-LINGO-1 antibody may only represent purely remyelination potential since the Mi group [117] has not provided any indirect or direct evidence for the limitation of axonal degeneration during the course of EAE. This potentially explains the failure of Opicinumab not meeting its endpoints in phase II SYNERGY trial (NCT01864148) in active relapsing-remitting and secondary progressive MS patients.

The delivery of NgR(310)ecto-Fc by HSCs and potential targeting of the NgR1 signaling operative only at inflammatory demyelinating lesions may modify the pathological environment, and subsequently enhance myelin debris clearance by macrophages. It is suggested that insufficient myelin debris clearance, prevalent in several neurodegenerative diseases, may be associated with an inadequate regenerative response [118]. Numerous studies in animal models of CNS and peripheral nervous system (PNS) injury have suggested faster clearance of myelin debris by macrophages potentially enhancing remyelination [118,119,120,121,122]. Furthermore, in an optic nerve injury model, deficient microglia activation has shown a reduced clearance of myelin debris [119]. It has been demonstrated that repeated systemic administration of the inflammatory agent, LPS, into a mouse model of SCI may stimulate the recruitment and increase the number of activated macrophages and microglia in the degenerating mouse dorsal column by upregulating CD14, a receptor for LPS. Furthermore, the phagocytosis of degenerative myelin was demonstrated by a significant reduction in the density of Luxol Fast Blue (LFB) staining and conversely a substantial increase in the number of round phagocytic macrophages and levels of Oil Red O (ORO) staining for phagocytosed myelin debris, indicating rapid clearance of degenerating myelin at the site of injury [123]. Further evidence of enhanced myelin clearance by macrophages culminating in remyelination may be supported by studies within the PNS whereby promoting macrophage entrance and phagocytic activity in degenerating nerve fibers in the PNS can repair myelin [120]. On the other hand, the reduced efficiency of macrophage accumulation in the injured sciatic nerve, impaired clearance of myelin debris, and decreased axonal remyelination leading to delayed regeneration [121]. Furthermore, the regeneration exhibited in the PNS was achieved possibly due to rapid clearance of myelin debris after Wallerian degeneration (WD), which could be applied to the sequelae of CNS regeneration [122].

## 7. Conclusions

Permanent neurological disability in MS patients is governed by axonal damage and demyelination within the CNS, a consequence of neurodegeneration subsequent to the burden of inflammatory damage over time and currently an unmet medical need. Current therapeutics in MS target the inflammatory nature of the disease to limit the autoimmune attack on CNS myelin that include, depleting B cells, preventing transendothelial cell immune migration and infiltration, or halting the egress of lymphocytes from the lymphoid tissues. Attempts at driving repair of the central nervous system following acute or chronic demyelination are generating substantial knowledge from animal models but achieving the appropriate translation of such therapeutics are currently outside of being a clinical reality. One particular regenerative candidate could be the NgR(310)ecto-Fc decoy protein that antagonizes NgR1-dependent signaling in the central nervous system and has shown regenerative potential in the animal models of various CNS disorders, that may include progressive MS. However, achieving an expedited clearance of myelin debris near or within the demyelinated lesions is what is required if we are to modify the CNS environment for repair. Delivering this NgR(310)ecto-Fc decoy protein directly to lesions to clear them for myelin debris may occur through the utility of autologously transplanting genetically modified HSCs that can lineage differentiate toward immune cells that are attracted to active MS lesions. This approach may be a more robust targeted means to deliver regenerative therapeutics for MS patients who are not responsive to any conventional therapy and may also hold great therapeutic promise for other neurodegenerative disorders.

## Figures and Tables

**Figure 1 cells-08-00001-f001:**
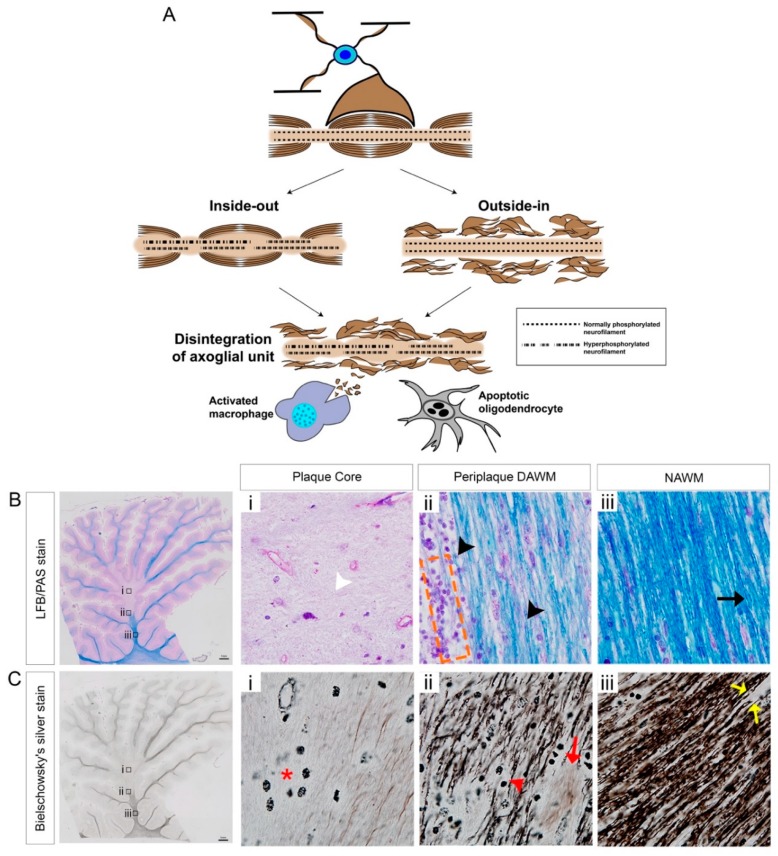
Schematic representation of ‘inside-out’ and ‘outside-in’ models which may govern axo-glial disintegration, leading to accumulated myelin debris in chronic-active MS lesions. (**A**) Myelinated axons are supported by healthy oligodendrocytes and consist of normally phosphorylated neurofilaments for appropriate neuronal architecture. The ‘inside-out’ paradigm (on the left-hand side) is exhibited as a primary axonal degeneration along with hyperphosphorylated neurofilaments with an intact myelin sheath, whereas the ‘outside-in’ paradigm (on the right-hand side) is characterized as the primary demyelination with intact axons. Both paradigms can lead to axo-glial disintegration and oligodendrocyte death. Subsequently, activated macrophages and microglia phagocytose the stripped myelin and extracellular myelin debris. (**B**) Luxol fast blue combined with periodic acid Schiff (LFB-PAS) stained chronic-active MS brain lesions from a secondary progressive MS patient; (**B-i**) plaque core center illustrating the hypocellular or paucicellular nature of the lesion (indicated by white arrowhead), (**B-ii**) periplaque dirty-appearing white matter (DAWM) demonstrating the degenerated myelin (indicated by black arrowheads) along with the hypercellular immune infiltrates (orange dashed box), (**B-iii**) normal-appearing white matter (NAWM) demonstrating even myelin LFB/PAS staining (dark blue, indicated by black arrow). (**C**) Bielschowsky silver stained chronic-active MS brain lesion; (**C-i**) plaque core demonstrates axonal dropout and macrophages (indicated by red asterisk) along with occasional dystrophic axons (**C-ii**) periplaque demyelinated active white matter (DAWM) demonstrating retraction bulbs (indicated by red arrowhead) along with transected axons (indicated by red arrow), (**C-iii**) normal-appearing white matter (NAWM) demonstrating transected axons (indicated by yellow arrow). Scale bar = 1 mm.

**Figure 2 cells-08-00001-f002:**
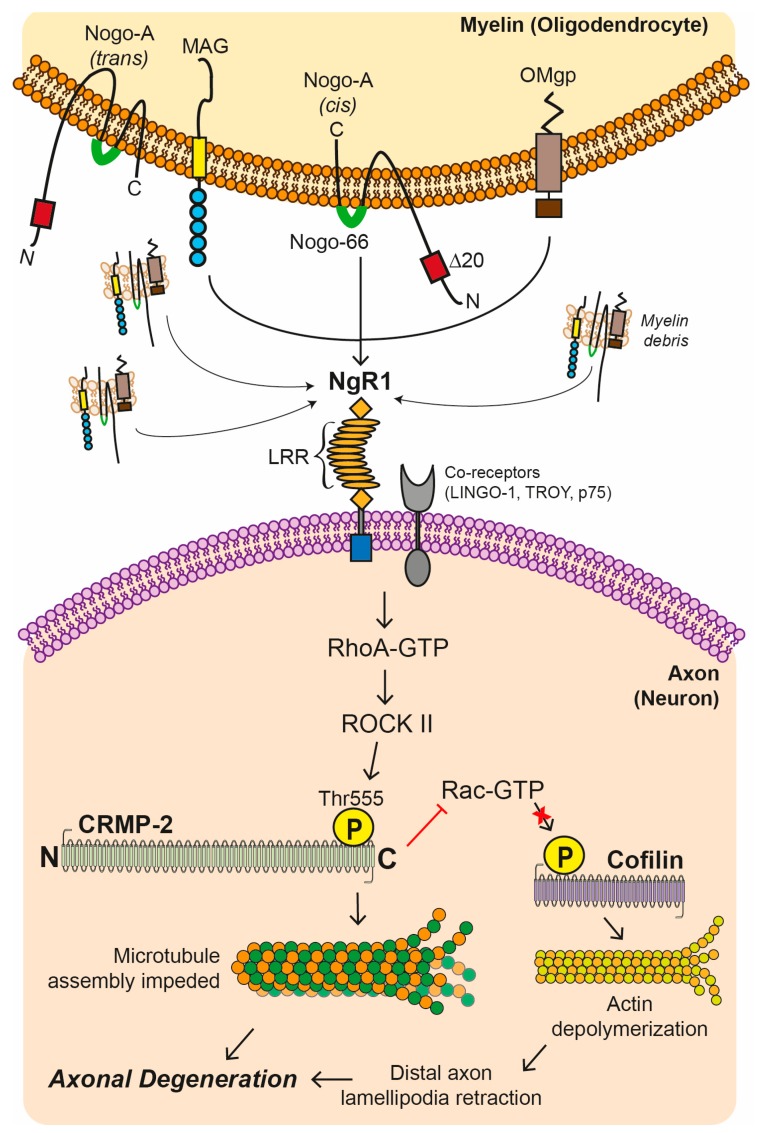
Nogo receptor 1 (NgR1)-dependent signaling cascade.

**Figure 3 cells-08-00001-f003:**
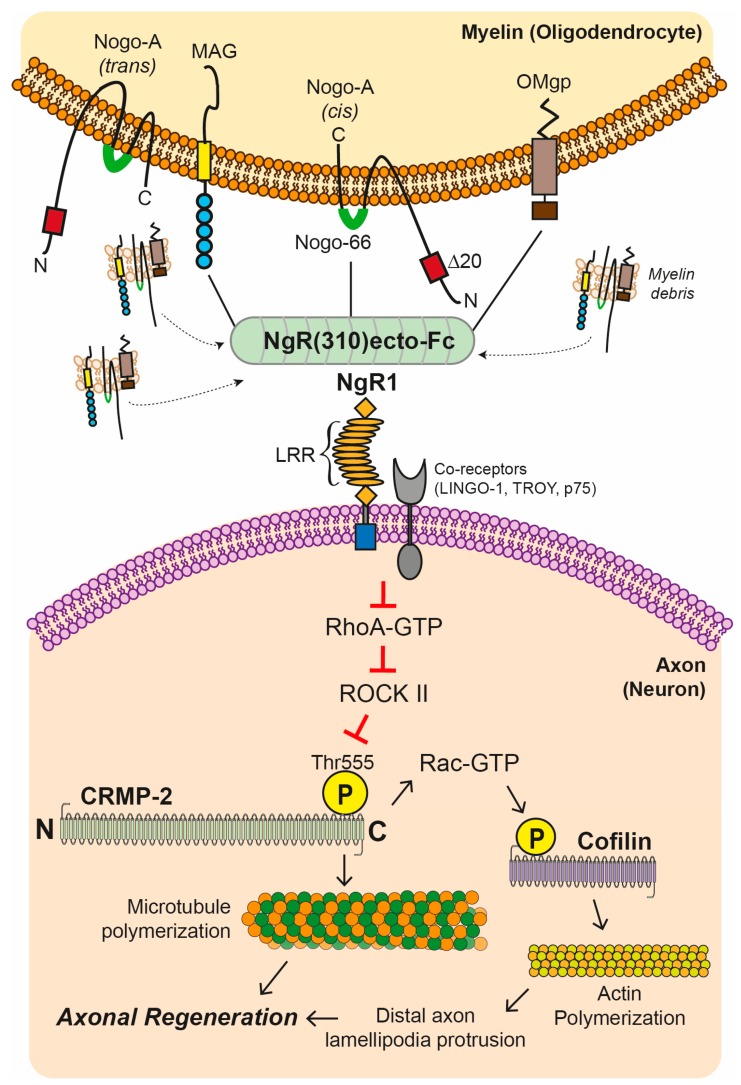
Blockade of Nogo receptor 1 (NgR1)-dependent signaling cascade with NgR(310)ecto-Fc fusion protein.

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
