# Peer review of "Can We Design a Nogo Receptor-Dependent Cellular Therapy to Target MS?"

_cells, 2018, doi:10.3390/cells8010001_

Round 1

Reviewer 1 Report

This review article is an interesting and informative set of considerations, and the authors add to our knowledge of how we can design and approach to the future possible therapy for multiple sclerosis. The manuscript is well documented, however, I think it might be better to add describing of actin assembly by Nogo signaling and also other methodology for Nogo signaling suppression. Some important comments are as follows. Please address them for revision.

1)    The authors describe that RhoA/ROCK downstream of Nogo signaling causes only CRMP2 phosphorylation (p9, lines 296-307, and Figs 2-3). However, not only change in microtubule assembly but also actin assembly is well known as Nogo signaling-induced event. It might be better that the authors add Nogo signaling-induced actin depolymerization in addition to CRMP2 phosphorylation and show both of signaling in figures.

2)    The authors mentioned NgR(310)ecto-FC as a potential therapeutic intervention as the most important methodology. For the benefit of the readers, I think it might be better to add other potentials of therapeutic approach such as anti-NgR antibody, anti-Nogo antibody, Nogo antagonist NEP1-40 and endogenous NgR antagonist LOTUS protein.

3)    Finally, many reference numbers are incorrect. The authors should check the number cited very carefully.

Author Response

This review article is an interesting and informative set of considerations, and the authors add to our knowledge of how we can design and approach to the future possible therapy for multiple sclerosis. The manuscript is well documented, however, I think it might be better to add describing of actin assembly by Nogo signaling and also other methodology for Nogo signaling suppression. Some important comments are as follows. Please address them for revision.

1)     The authors describe that RhoA/ROCK downstream of Nogo signaling causes only CRMP2 phosphorylation (p9, lines 296-307, and Figs 2-3). However, not only change in microtubule assembly but also actin assembly is well known as Nogo signaling-induced event. It might be better that the authors add Nogo signaling-induced actin depolymerization in addition to CRMP2 phosphorylation and show both of signaling in figures.

Answer (A): This has now been amended in Figures 2 and 3.

2)     The authors mentioned NgR(310)ecto-FC as a potential therapeutic intervention as the most important methodology. For the benefit of the readers, I think it might be better to add other potentials of therapeutic approach such as anti-NgR antibody, anti-Nogo antibody, Nogo antagonist NEP1-40 and endogenous NgR antagonist LOTUS protein.

(A) The title is actually “designing NgR-dependent cellular therapies” whereby the other therapies stipulated by the reviewer have never been designed and utilised in MS or its models of disease (except for anti-Nogo A antibody therapy which has been discussed on pg 12). The effect of the soluble LOTUS protein and its capacity to modulate NgR-dependent signalling in MS is now defined on pg14-15.

3)    Finally, many reference numbers are incorrect. The authors should check the number cited very carefully.

(A) All references have now been amended due to endnote errors.

Reviewer 2 Report

I commend the authors for the detailed but still concisely written review on Nogo-A-based approaches as potential remyelinating therapies for MS. As the authors state, finding remyelinating therapies is an unmet and urgend medical need, especially as disability in MS patients could recover to a certain amount upon successful re-wrapping of axons. Also, the figures of the review are appealing and comrehendable.

Whereas I like this review, I have a couple of comments to address for the authors:

1.) In the paragraph regarding abolishment of axonal transport during early axonal damage, it might be appropriate to also briefly elaborate on the work of M. Kerschensteiner, e.g. Sorbara et al., Pervasive axonal transport deficits in multiple sclerosis models, Neuron, 2014. Kerschensteiners group has done many interesting and important studies in this regard (even though the evidence mainly comes from animal models) and I think it would be worthwile mentioning.

2.) Whereas the M1/M2 concept is still widely accepted, its correctness has recently been debated (e.g Martinez and Gordon, The M1 and M2 paradigm of macrophage activation: time for reassessment, F1000, 2014 or Ransohoff, A polarizing question: do M1 and M2 microglia exist? Nat Neurosci. 2016). Thus, I It think it would be appropriate to balance the discussion on this nevertheless interesting issue.

3.) Line 238/239: The references 37 and 38 might be not the correct ones as these studies do not deal with EAE or MS. It think there is a general issue with your references as ref 50 is also not correct.

4.) Paragraph 6: the discussion of NgR ecto, HSCDT and Lingo-ab for potential remyelinating therapies seems arbitrary. There are numerous other remyelinating approaches, some of which already entered clinical trials. I think it would be legitimate to discuss Nogo-A-based strategies, but then, HSCDT would not fit into this paragraph. Why did you choose these therapies in particular?

5.) You nicely elaborate on the inside-out versus outside-in hypothesis. For me, one pivotal piece of evidence which you still could add to this discussion is the whole story around metabolic deprivation of axons upon demyelination (e.g. Glycolytic oligodendrocytes maintain myelin and long-term axonal integrity. Funfschilling et al., Nature, 2012 and/or others).

6.) Minor comments:

Line 381: OLYMPUS instead of OLYLMPUS.

The text flow seems a bit odd from sentence starting line 391 and sentence starting line 392.

Author Response

I commend the authors for the detailed but still concisely written review on Nogo-A-based approaches as potential remyelinating therapies for MS. As the authors state, finding remyelinating therapies is an unmet and urgend medical need, especially as disability in MS patients could recover to a certain amount upon successful re-wrapping of axons. Also, the figures of the review are appealing and comrehendable.

Whereas I like this review, I have a couple of comments to address for the authors:

1.)  In the paragraph regarding abolishment of axonal transport during early axonal damage, it might be appropriate to also briefly elaborate on the work of M. Kerschensteiner, e.g. Sorbara et al., Pervasive axonal transport deficits in multiple sclerosis models, Neuron, 2014. Kerschensteiners group has done many interesting and important studies in this regard (even though the evidence mainly comes from animal models) and I think it would be worthwile mentioning.

(A) This has now been included on pg16-17.

2.)            Whereas the M1/M2 concept is still widely accepted, its correctness has recently been debated (e.g Martinez and Gordon, The M1 and M2 paradigm of macrophage activation: time for reassessment, F1000, 2014 or Ransohoff, A polarizing question: do M1 and M2 microglia exist? Nat Neurosci. 2016). Thus, I It think it would be appropriate to balance the discussion on this nevertheless interesting issue.

(A) This has now been included on pg7 paragraph 1.

3.)            Line 238/239: The references 37 and 38 might be not the correct ones as these studies do not deal with EAE or MS. It think there is a general issue with your references as ref 50 is also not correct.

(A) All references have now been checked and amended and occurred due to end note errors.

4.)            Paragraph 6: the discussion of NgR ecto, HSCDT and Lingo-ab for potential remyelinating therapies seems arbitrary. There are numerous other remyelinating approaches, some of which already entered clinical trials. I think it would be legitimate to discuss Nogo-A-based strategies, but then, HSCDT would not fit into this paragraph. Why did you choose these therapies in particular?

(A) The current review discusses the future of targeted approaches for neuroprotection and neurorepair thereby limiting off-target effects. HSCT approaches is initially described to introduce the current clinical landscape for cellular therapies that can be adapted for anti-Nogo A dependent strategies that may be more specific to modify the chronic MS lesion microenvironment. These transplanted cells can lineage differentiate to mature immune cells that hone in on active lesion areas. This is very pertinent and sets the scene for our recent novel paradigm that is planned for publication next year.

5.) You nicely elaborate on the inside-out versus outside-in hypothesis. For me, one pivotal piece of evidence which you still could add to this discussion is the whole story around metabolic deprivation of axons upon demyelination (e.g. Glycolytic oligodendrocytes maintain myelin and long-term axonal integrity. Funfschilling et al., Nature, 2012 and/or others).

 (A) This is a terrific suggestion and we have now included this on pg11.

6.) Minor comments:

Line 381: OLYMPUS instead of OLYLMPUS.

(A): This has now been amended

The text flow seems a bit odd from sentence starting line 391 and sentence starting line 392.

(A): This has now been amended

Round 2

Reviewer 2 Report

I congratulate the authors for this sound review. All my concerns have been addressed.